# Intestinal fibroblastic reticular cell niches control innate lymphoid cell homeostasis and function

Hung-Wei Cheng[1,3], Urs Mörbe [1,3], Mechthild Lütge [1], Céline Engetschwiler[1], Lucas Onder[1], Mario Novkovic [1], Cristina Gil-Cruz[1], Christian Perez-Shibayama[1], Thomas Hehlgans[2], Elke Scandella[1] & Burkhard Ludewig [1✉]

Innate lymphoid cells (ILCs) govern immune cell homeostasis in the intestine and protect the host against microbial pathogens. Various cell-intrinsic pathways have been identified that determine ILC development and differentiation. However, the cellular components that regulate ILC sustenance and function in the intestinal lamina propria are less known. Using single-cell transcriptomic analysis of lamina propria fibroblasts, we identify fibroblastic reticular cells (FRCs) that underpin cryptopatches (CPs) and isolated lymphoid follicles (ILFs). Genetic ablation of lymphotoxin-β receptor expression in *Ccl19*-expressing FRCs blocks the maturation of CPs into mature ILFs. Interactome analysis shows the major niche factors and processes underlying FRC-ILC crosstalk. In vivo validation confirms that a sustained lymphotoxin-driven feedforward loop of FRC activation including IL-7 generation is critical for the maintenance of functional ILC populations. In sum, our study indicates critical fibroblastic niches within the intestinal lamina propria that control ILC homeostasis and functionality and thereby secure protective gut immunity.

[1] Institute of Immunobiology, Kantonsspital St. Gallen, St. Gallen, Switzerland. [2] Leibniz Institute of Immunotherapy (LIT), Chair for Immunology, University of Regensburg, Regensburg, Germany. [3]These authors contributed equally: Hung-Wei Cheng, Urs Mörbe. ✉email: burkhard.ludewig@kssg.ch

The intestinal immune system harbors diverse immune cell populations that are instructed within discrete lymphoid structures to contain the microbiome and to regulate immune responses against food antigens, while staying alert and functional in response to the invasion by pathogens[1,2]. Innate lymphoid cells (ILCs) operate independently of specific antigen receptor signaling and are particularly abundant in the intestinal lamina propria[3]. ILCs can be grouped into functionally distinct subsets including natural killer cells with cytotoxic potential, ILC1 with T helper 1-like functions, T helper 2 cytokine-producing ILC2 and the most abundant ILC3 that are characterized by the expression of the transcription factor RORγt and secretion of anti-bacterial cytokines IL17 and IL22[4,5]. In spite of various phenotypes and functions of ILC subsets, all ILCs are generated from a common progenitor (ILCp) that is present in the fetal liver during embryogenesis or in the bone marrow of adult individuals[6,7]. Fetal ILCp seed tissues, expand and locally differentiate concomitantly with the microbial colonization of the host[8–10]. Adult ILCp in humans have been shown to migrate to different organs and contribute to local, tissue-resident ILC pools[11]. Parabiosis and cell fate-mapping experiments in mice have confirmed replenishment of ILCs from circulating ILCp or migratory mature ILCs and revealed increased ILCp and ILC recruitment during inflammation[12–14]. Regardless of their origin, i.e., coming from tissue-resident or recently immigrated ILCp, differentiated ILC populations acquire tissue-specific molecular traits that are imprinted by the local microenvironment[7,15–17]. It is therefore important to characterize the key niche-forming cells that support ILC maintenance and functionality.

ILCs are indispensable for the formation of gut-associated secondary lymphoid organs (SLOs) during embryogenesis[18,19]. Both mesenteric lymph nodes and Peyer's patches fail to form in the absence of embryonic ILC3, which are also known as RORγt+ lymphoid tissue inducer cells[20,21]. In contrast, solitary intestinal lymphoid tissue (SILT), i.e., cryptopatches (CPs) that are formed by clusters of RORγt+ cells, immature isolated lymphoid follicles (imILF) that appear as RORγt+ cell clusters mixed with few B cells, and mature isolated lymphoid follicles (mILFs) that are composed of large B-cell clusters surrounded by a corona of RORγt+ cells, are induced in the intestinal lamina propria after birth[22]. SILT number, size, and degree of differentiation are driven by the colonizing microbiota and food components[23,24]. Similar to the organogenesis of SLOs, the clustering of RORγt+ ILC3 and signaling via the lymphotoxin β-receptor (LTβR) axis are critical steps to initiate the CP formation[22]. Further differentiation and maturation of CPs into organized ILFs appears to be regulated by a multitude of factors including ILC-derived lymphotoxin (LT)[25], aryl hydrocarbon receptor-signal-dependent activation of ILC3[23,26] and a large range of bacterial products that are sensed by Toll-like receptors or the nucleotide-binding oligomerization domain receptors[27]. However, the hierarchy of signals that determine CP to ILF transition and the processes that steer ILC communication with the cells that underpin the SILT infrastructure have remained largely unknown.

Fibroblastic reticular cells (FRCs) are the key cells that determine SLO compartmentalization and generate specialized niches for immune cell interactions[28–30]. Here, we provide a detailed molecular characterization of SILT FRCs and elaborate on the mechanisms underlying ILC homeostasis and function. Single-cell transcriptomic analysis of intestinal fibroblasts identifies *Ccl19*-expressing cells as SILT FRCs. Genetic ablation of lymphotoxin-β receptor (*Ltbr*) expression in Ccl19-Cre+ cells shows a crucial function of SILT FRCs for the transition of CPs to mILFs. The continued activation of SILT FRCs is required for the maintenance of effective ILC pools in the lamina propria and for protection against *Citrobacter rodentium* infection. These results

further characterize the nature of highly specialized SILT FRCs and document the molecular ILC-FRC interactome that is required for the establishment of intestinal immunity.

## Results

**Molecular characterization of SILT-underpinning FRCs.** To define potential fibroblastic ILC niches, we performed a high-resolution single-cell RNA-sequencing (scRNA-seq) analysis of lamina propria fibroblasts and identified nine clusters using uniform manifold approximation and projection (UMAP) (Fig. 1a and Supplementary Fig. 1a). At the chosen clustering resolution, the analysis revealed four related fibroblast populations expressing *Cd34* and *Pdgfra* that can be separated by the expression of *Cd81* (Fig. 1a) and correspond to the previously identified *Pdgfra*lo fibroblast and *Cd81*+ trophocyte populations, respectively[31]. The *Pdgfra*hi *Cd34*lo cluster with tenascin C (*Tnc*) expression represents telocytes that are situated mainly at the tip of the villi (Fig.1a and Supplementary Fig. 1b). The mRNAs encoding for a set of bone morphogenic proteins (*Bmp3*, *Bmp4*, *Bmp5*, *Bmp6,* and *Bmp7*) were highly enriched in the telocyte cluster, while mRNAs encoding for the BMP antagonist *Grem1* and epithelial niche factors such as R-Spondins (*Rspo2* and *Rspo3*) and Wnt (*Wnt2b* and *Wnt5a*) were enriched in the *Cd81*+ trophocytes and the *Pdgfra*lo cells (Supplementary Fig. 1a, left panel). Two *Acta2*hi *Cd34*− cell clusters could be detected that differed in *Ncam1* expression (Fig. 1a). The expression of perivascular cell markers (*Pdgfrb*, *Rgs5,* and *Esam*) (Supplementary Fig. 1a, middle panel) and co-staining for ACTA2 and NCAM1 in confocal microscopy analysis identified *Acta2*hi *Ncam1*− cells as mural cells situated around blood vessels (Supplementary Fig. 1c). High expression of molecules mediating cell contractility (*Tagln*, *Actg2,* and *Myh11*) marked *Ncam1*+ muscularis mucosae myocytes (Supplementary Fig. 1a, middle panel). The scRNA-seq analysis revealed a population of *Thy1*+ *Cd34*+ fibroblasts that were situated in the perivascular space close to the muscular layer (Supplementary Fig. 1d). The scRNA-seq analysis further identified a small cluster of cells expressing the chemokine *Ccl19*, the cytokine *Il7* (Fig. 1a), and an array of molecules that signify SLO FRCs (i.e., chemokines (*Cxcl16*, *Cxcl13,* and *Cxcl9*), cytokines (*Tnfsf13b*) and niche factors (*Il33* and *Kitl*)) (Supplementary Fig. 1a, right panel and Supplementary Data 1). The high expression of *Clu* in this population (Fig. 1a) facilitated the localization of clusterin (CLU)-positive cells in SILT structures (Supplementary Fig. 1e). The unique localization and the similarity of *Ccl19*+ *Clu*+ cells to SLO fibroblasts[32,33] suggested that the *Ccl19*-expressing lamina propria fibroblast population represents SILT FRCs.

Since the Ccl19-Cre transgene highlight FRCs in all SLOs[32,34,35], the Ccl19-Cre strain was crossed with R26R-EYFP mice (abbreviated as Ccl19-EYFP) to highlight lineage-traced cells in the lamina propria of the small intestine. Confocal microscopy revealed colocalization of EYFP+ cells with RORC+ ILC3 in SILTs (Fig. 1b). The high-resolution analysis identified EYFP+ SILT FRCs in CPs, immature and mILFs (Fig. 1c). EYFP+ FRCs appeared in developing SILTs around the time of weaning, i.e., in 3-week-old mice (Supplementary Fig. 2a). mILFs were not present directly after weaning and had developed the typical structure, i.e., large B-cell aggregations surrounded by a corona of RORC+ ILC3, when the mice had reached the age of 6 weeks (Supplementary Fig. 2a, b). Further in situ characterization by high-resolution confocal imaging confirmed that SILT FRCs are ACTA2low (Fig. 1d), NCAM1low (Fig. 1e), but show co-expression of EYFP and CLU (Fig. 1f). Based on the scRNA-seq data and the validation by confocal microscopy, we established a flow cytometry antibody panel and a gating strategy

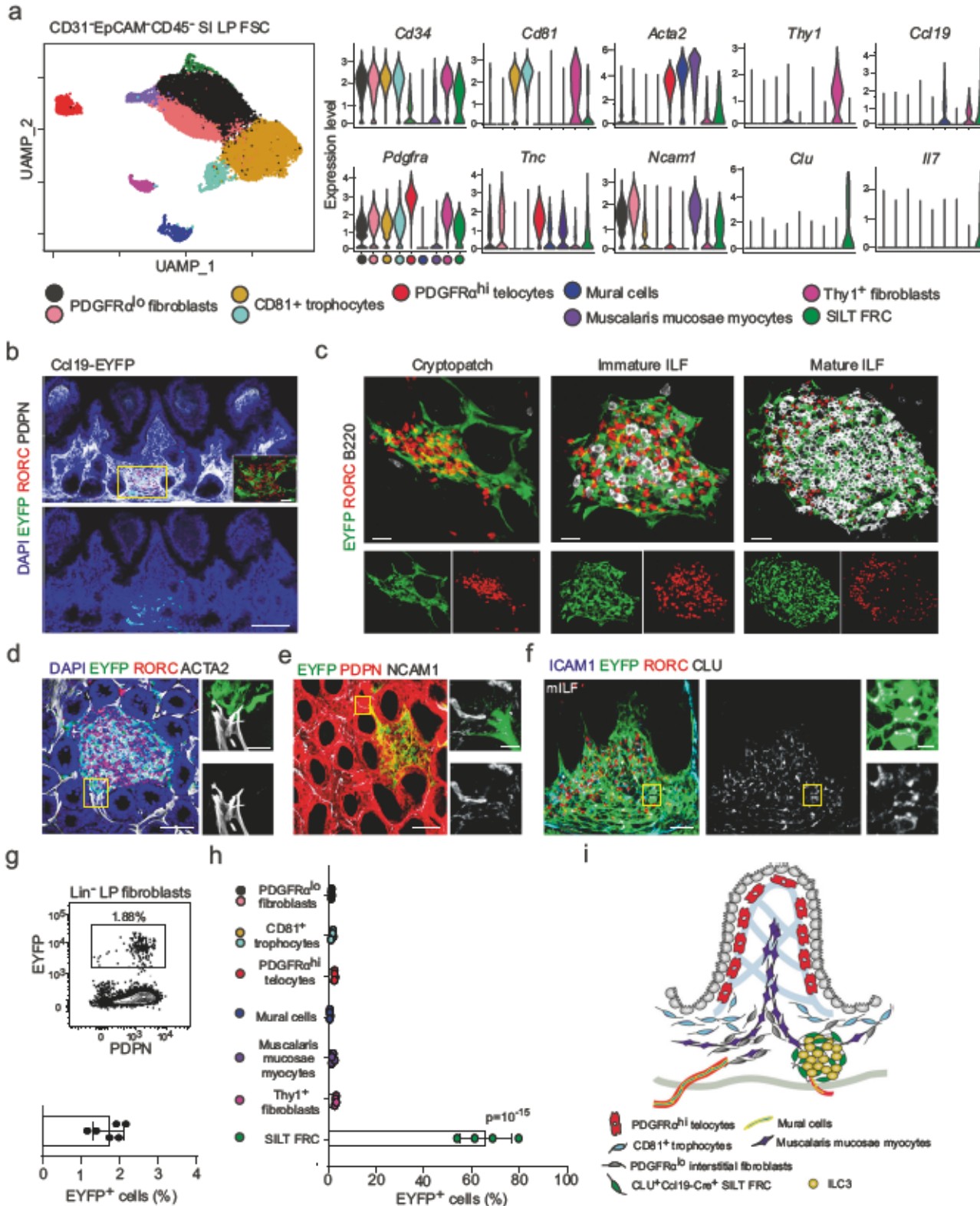

to distinguish the different lamina propria fibroblast populations (Supplementary Fig. 2c). Consistent with the small fraction of SILT FRCs in the lamina propria fibroblast landscape, we found less than 2% EYFP+ cells in the lineage-negative lamina propria cells of Ccl19-EYFP mice (Fig. 1g). As expected, EYFP-positive cells were almost exclusively confined to the CLU+ FRC population (Fig. 1h and Supplementary Fig. 2d). Taken together,

these results reveal the molecular identity of SILT FRCs within the heterogeneous intestinal fibroblast landscape (Fig. 1i).

**FRCs control SILT remodeling and maturation.** Signaling via the LTβR represents the first key signal for FRC differentiation and activation in lymph nodes[34], the splenic white pulp[32] and in Peyer's patches[35]. To assess to what extent LTβR-mediated

**Fig. 1 Identification of SILT-underpinning FRCs in the small intestine. a** UMAP of different intestinal lamina propria fibroblast populations from adult mice sorted for CD31⁻EpCAM⁻CD45⁻ cells. Right panels indicate the gene expression profiles in different clusters shown as violin plots. **b** Representative immunofluorescence images of small intestinal sections from adult Ccl19-EYFP mice stained with the indicated antibodies, Scale bar, 100 µm. Boxed area in (**b**) shows magnified SILTs underpinned by EYFP⁺ FRCs. Scale bar, 20 µm. **c** Representative images of SILT structures from adult Ccl19-EYFP stained with the indicated antibodies. Scale bars, 20, 20, and 40 µm in cryptopatch, imILF and mILF, respectively. **d–f** Representative immunofluorescence images of small intestinal sections from adult Ccl19-EYFP mice. Scale bars, 50 µm (**d**), 30 µm (**e**), and 30 µm (**f**). Boxed areas in (**d–f**) indicate the magnified regions shown in the right panels. Scale bars, 10 µm (**d**), 5 µm (**e**), and 5 µm (**f**). **g** Representative flow cytometric analysis of Ccl19-EYFP transgene-targeted cells and bar graphs showing the percentage of EYFP⁺ cells. **h** Percentage of EYFP⁺ cells in lamina propria fibroblast populations in the small intestine, based on the gating strategy in Supplementary Fig. 2c, d. **i** Schematic description of the spatial distribution of lamina propria fibroblasts in the mouse small intestine. **a** ScRNA-seq data represents 25,614 lamina propria intestinal fibroblastic stromal cells, $n = 4$ biological replicates. **b–f** Images are representative of at least four mice. **g** $n = 6$ mice from two independent experiments, mean ± SEM. **h** $n = 4$ mice from two independent experiments, mean ± SEM. Statistical analysis was performed using one-way ANOVA with Tukey's multiple comparison test.

activation of FRCs affects SILT formation, we crossed Ccl19-EYFP mice to $Ltbr^{fl/fl}$ mice (Ccl19-EYFP $Ltbr^{fl/fl}$). Whereas CPs and imILFs were still formed in the absence of LTβR signaling in SILT FRCs (Fig. 2a), the formation of mILFs was completely abrogated in Ccl19-EYFP $Ltbr^{fl/fl}$ mice (Fig. 2b). Quantitative image analysis revealed that the formation of imILFs was significantly reduced by the genetic ablation of $Ltbr$ expression in SILT FRCs, both in frequency (Fig. 2b) and in size (Fig. 2c). In contrast to $Ltbr$-competent SILT FRCs, which form a dense cellular network (Fig. 1c), $Ltbr$-deficient EYFP⁺ SILT FRCs appeared as poorly branched cells situated in the periphery of CPs and imILFs (Fig. 2a). To decipher the molecular traits of SILT FRC in the absence of LTβR signaling, we sorted EYFP⁺ cells from the small intestine of Ccl19-EYFP and Ccl19-EYFP $Ltbr^{fl/fl}$ mice and analyzed the cells by scRNA-seq analysis (Fig. 2d). Differentially expressed genes in SILT FRCs under conditions of $Ltbr$-deficiency included niche factors such as $Il7$, $Flt3l$, the chemokines $Ccl19$, $Ccl21a$, $Cxcl13$, and other FRC markers such as $Mfge8$ and $Clu$ (Fig. 2e and Supplementary Fig. 3a). The expression of other genes that are expressed by SLO FRCs, such as $Cxcl16$, $Ccl2$, $Kitl$ or $Il33$, was LTβR-independent in SILT FRCs (Fig. 2e). The LTβR-dependent activation pattern of SILT FRCs was confirmed by RT-PCR analysis using sorted PDPN⁺ EYFP⁺ and PDPN⁺ EYFP⁻ cells from the lamina propria of Ccl19-EYFP and Ccl19-EYFP $Ltbr^{fl/fl}$ mice (Supplementary Fig. 3b). As expected, the expression of $Il7$, $Clu$, and $Ccl19$ was significantly higher in $Ltbr$-proficient SILT FRCs when compared to $Ltbr$-deficient SILT FRCs or $Ltbr$-proficient non-SILT fibroblasts (Fig. 2f and Supplementary Fig. 3c). The RT-PCR analysis also confirmed the LTβR-independent regulation of the alarmin $Il33$ (Supplementary Fig. 3c). High-resolution confocal microscopy further confirmed that CLU expression by FRCs is LTβR-dependent in all SILT maturation stages (Fig. 2g) and that the B-cell-attracting factor CXCL13 is lacking in the absence of LTβR signaling in SILT FRCs (Fig. 2h). These data indicate that SILT remodeling and maturation depend exclusively on the appropriate activation and differentiation of SILT FRCs.

**SILT FRCs govern ILC homeostasis and functionality.** To assess to what extent the block of SILT maturation affects immune cell homeostasis in the small intestine, we evaluated the cellular composition in the lamina propria from Ccl19-Cre $Ltbr^{fl/fl}$ mice in comparison to co-housed Cre-negative littermates. We found that neither T- nor B-cell populations (Supplementary Fig. 4a, b) nor myeloid cell subsets (Supplementary Fig. 4c, d) were affected by the $Ltbr$-deficiency in SILT FRCs. Whereas the relative composition of ILC subsets (i.e., Tbet⁺ RORC⁻ ILC1, GATA3⁺ RORC⁻ ILC2, and GATA3⁻ RORC⁺ ILC3) was maintained despite the $Ltbr$-deficiency in SILT FRCs (Fig. 3a and Supplementary Fig. 5a, b), the fraction and absolute numbers of Lin⁻ CD127⁺ ILCs were

significantly reduced in the Ccl19-Cre $Ltbr^{fl/fl}$ mice (Fig. 3b). In particular, ILC1 and ILC3 subsets showed a significant reduction in cell numbers in the lamina propria under conditions of $Ltbr$-deficiency in Ccl19-Cre⁺ cells (Fig. 3c and Supplementary Fig. 5b). Assessment of ILC proliferation and cell death markers revealed that $Ltbr$-deficiency in SILT FRCs mainly affected the survival of ILC3 as shown by the significantly elevated expression of active caspase 3/7 in the GATA3⁻ RORC⁺ subset (Supplementary Fig. 5c, d). These data indicate that FRC-mediated SILT maturation is crucial for the maintenance of ILC pools in the intestinal lamina propria.

Next, we assessed the functional consequences of $Ltbr$-deficiency in SILT FRCs and found significantly reduced numbers of IL17- or IL22-secreting ILC3 in the lamina propria of Ccl19-Cre $Ltbr^{fl/fl}$ mice in comparison to co-housed Cre-negative littermates (Fig. 3d and Supplementary Fig. 5e). Since LT-signaling modulates the IL22 production by ILCs[36] and IL22 induces the production of antimicrobial peptides in intestinal epithelial cells[9,37], we examined the expression of genes encoding for antimicrobial peptides. We found significantly reduced expression of $RegIIIg$, $S100a8$, and $S100a9$ in the small intestine of Ccl19-Cre $Ltbr^{fl/fl}$ mice (Fig. 3e). Mice with $Ltbr$-deficient SILT FRCs showed reduced numbers of functional ILC3 and diminished expression of antimicrobial peptides that most likely led to the increased susceptibility to infection with the bacterial pathogen $C. rodentium$ (Fig. 3f). Impaired pathogen control in Ccl19-Cre $Ltbr^{fl/fl}$ mice resulted in significantly reduced weight gain (Fig. 3g), reduced colon length (Fig. 3h), and increased inflammation (Fig. 3i) when compared to co-housed Cre-negative littermates. These data support the notion that SILT FRCs generate microenvironmental niches that are crucial for the maintenance of functional ILC populations to secure efficient control of intestinal pathogens.

**Persistent SILT FRC-ILC crosstalk sustains intestinal immunity.** To assess whether sustained communication between FRCs and ILCs is required for the maintenance of mature SILT structures and resistance to pathogen challenge, we utilized the Ccl19-tTA inducible mouse model[32] crossed with R26R-EYFP and $Ltbr^{fl/fl}$ mice (abbreviated as Ccl19-iEYFP $Ltbr^{fl/fl}$) (Supplementary Fig. 6a). Due to the expression of the tetracycline transactivator in $Ccl19$-expressing cells, Cre recombinase activity can be regulated through the provision of doxycycline[32]. Here, we used the non-antibiotic doxycycline derivative 4-epidoxycline (abbreviated as Dox)[38] to concomitantly regulate the expression of the EYFP marker and ablation of $Ltbr$ expression by first keeping pregnant dams and the offspring on Dox until the age of 8 weeks and subsequent withdrawal for 2 or 8 weeks (Fig. 4a). In the absence of Dox treatment, the Ccl19-iEYFP model faithfully recapitulated the phenotype of the Ccl19-EYFP mouse model with genetic targeting of FRCs underpinning all SILT stages

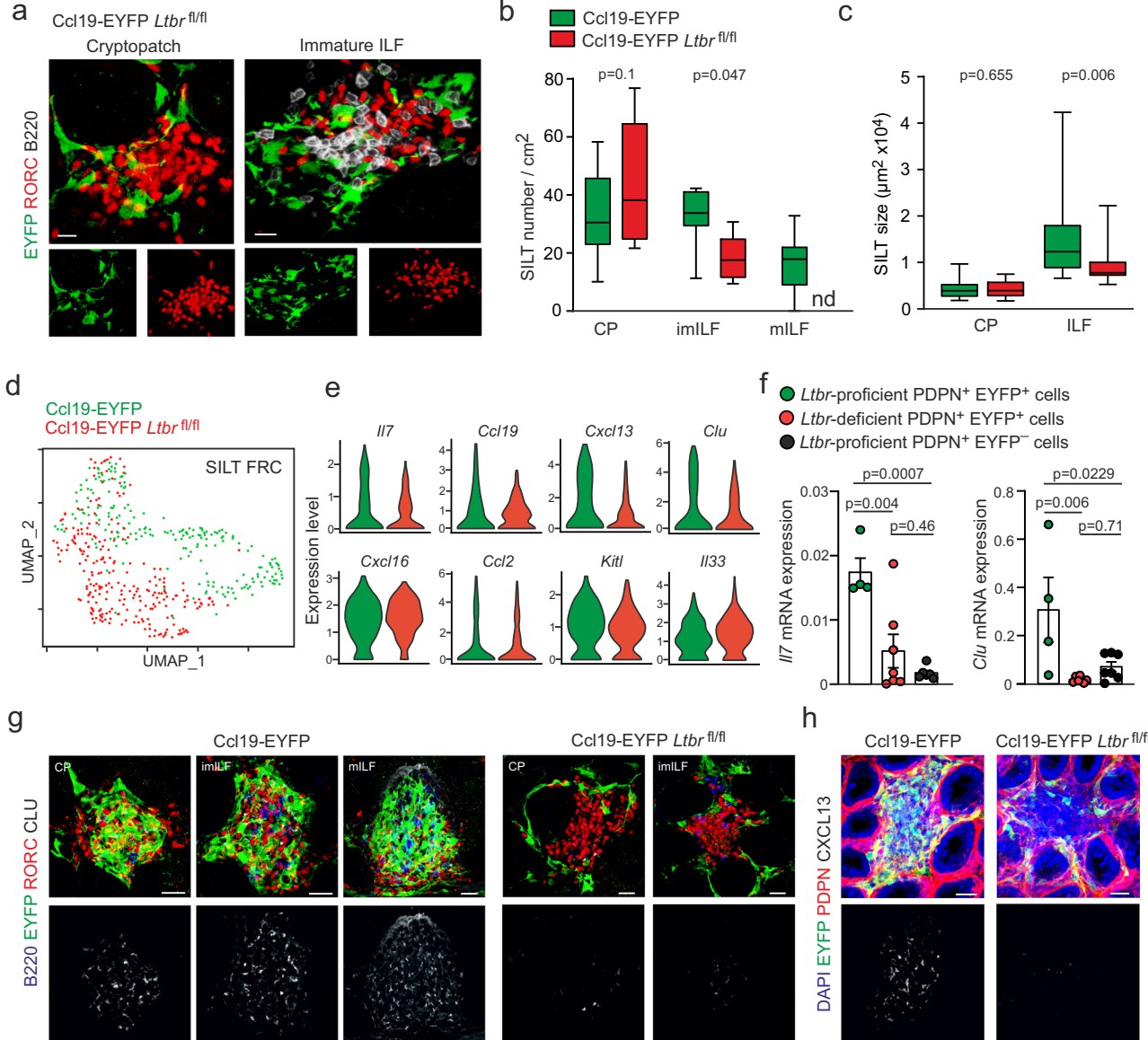

**Fig. 2 FRC activation controls SILT maturation. a** Representative confocal microscopy images of SILTs from adult Ccl19-EYFP $Ltbr^{fl/fl}$ mice stained with the indicated antibodies. Scale bar, 20 µm. **b** Number of SILTs detected in the small intestine from adult Ccl19-iEYFP and Ccl19-EYFP $Ltbr^{fl/fl}$ mice. **c** Average SILT size in the small intestine of adult Ccl19-iEYFP and Ccl19-EYFP $Ltbr^{fl/fl}$ mice **d** UMAP of re-embedded SILT FRC from sorted EYFP+ cells of adult Ccl19-iEYFP and Ccl19-EYFP $Ltbr^{fl/fl}$ mice. **e** Violin plots of the indicated gene expression profiles in different conditions based on scRNA-seq readout. **f** Relative expression of $Il7$ and $Clu$ in FACS sorted PDPN+ EYFP+ SILT FRC and PDPN+ EYFP- lamina propria fibroblasts from Ccl19-EYFP and Ccl19-EYFP $Ltbr^{fl/fl}$ small intestine, measured by real-time PCR. **g, h** Small intestinal sections from adult Ccl19-EYFP and Ccl19-EYFP $Ltbr^{fl/fl}$ mice analyzed by confocal microscopy after staining with the indicated antibodies. Scale bar, 20 µm in (**g, h**). **a** Images are representative of at least six mice. **b** $n = 8$ and 8 mice from three independent experiments, median ± interquartile ranges. **c** $n = 32$ and 32 in CP and $n = 37$ and 13 in ILF from 8 mice in each condition with three independent experiments, median ± interquartile ranges. **d, e** ScRNA-seq data represents EYFP+ SILT FRCs from adult Ccl19-EYFP and Ccl19-EYFP $Ltbr^{fl/fl}$ mice, $n = 4$ biological replicates. **f** $n = 4$, 7, and 7 from Ccl19-EYFP and Ccl19-EYFP $Ltbr^{fl/fl}$ mice, respectively, from three independent experiments, mean ± SEM. **g, h** Images are representative of at least four mice. Statistical analyses were performed using non-parametric two-tailed Mann–Whitney test (**c**) and one-way ANOVA with Tukey's multiple comparison test (**b, f**).

(Supplementary Fig. 6b). As expected, the formation of SILTs was impaired in Ccl19-iEYFP $Ltbr^{fl/fl}$ mice in the absence of Dox treatment, i.e., with active Cre recombinase expression (Fig. 4b, no Dox). Consistent with data from the Ccl19-Cre model[35], continued ablation of the $Ltbr$ gene in Ccl19-iEYFP $Ltbr^{fl/fl}$ mice (no Dox) led to reduced Peyer's patch formation (Supplementary Fig. 6c). Dox treatment of pregnant dams and postnatal provision of Dox via the drinking water (Dox on) prevented Cre-mediated $Ltbr$-deletion and EYFP-expression and hence allowed for SILT maturation and Peyer's patch formation

(Fig. 4b and Supplementary Fig. 6c, d). At the age of 8 weeks, all ILC subsets were reduced in untreated Ccl19-iEYFP $Ltbr^{fl/fl}$ mice compared to $Ltbr$-competent Ccl19-iEYFP mice (Supplementary Fig. 6e). Blocking of Cre-mediated $Ltbr$-ablation through Dox treatment restored ILC homeostasis in Ccl19-iEYFP $Ltbr^{fl/fl}$ mice (Supplementary Fig. 6e). To reactivate Cre-mediated recombination in Dox-treated Ccl19-iEYFP $Ltbr^{fl/fl}$ mice, we ceased the provision of Dox via the drinking water at the age of 8 weeks (Fig. 4a). After Dox removal for 2 weeks (Dox off 2 wk), Cre-mediated recombination had started and first EYFP+ cells

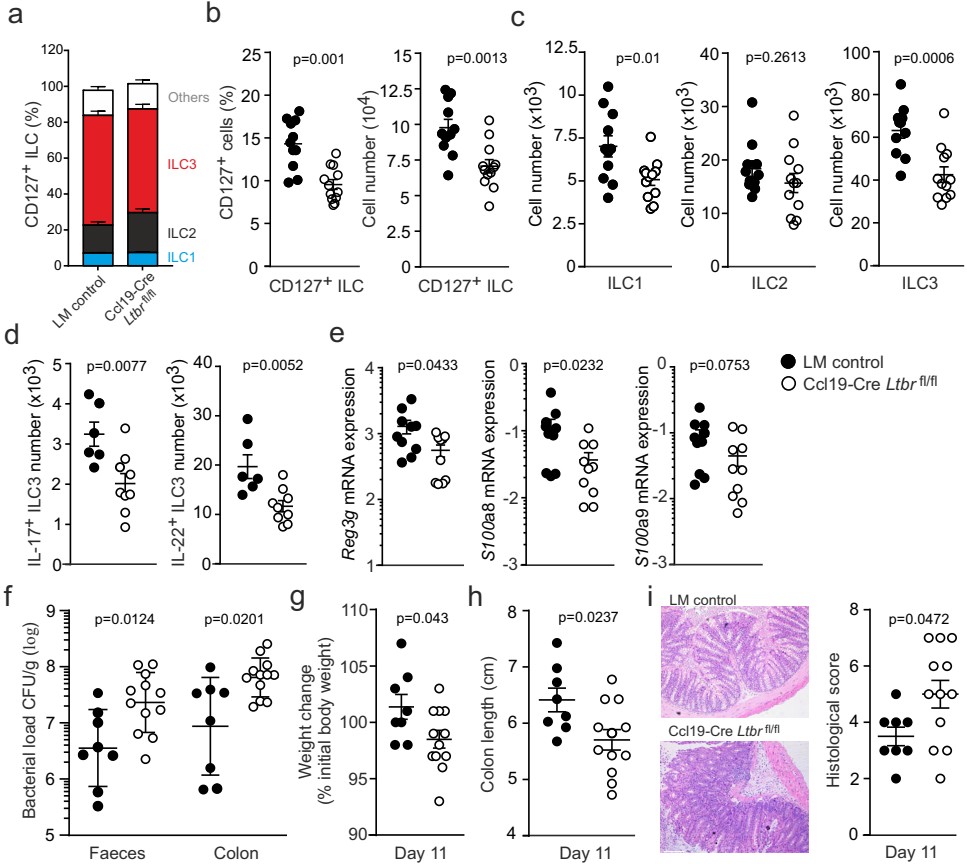

**Fig. 3 Innate lymphoid cell sustenance and intestinal immunity in the absence of LTβR signaling in SILT FRCs. a** Subset composition in CD127[+] ILCs isolated from adult Ccl19-Cre Ltbr[fl/fl] mice and co-housed littermate controls. **b** Percentage and cell numbers of CD127[+] ILCs and ILC subsets from adult Ccl19-Cre Ltbr[fl/fl] mice and co-housed littermate controls (LM control). **c** Quantification of ILC subsets from adult Ccl19-Cre Ltbr[fl/fl] mice and co-housed littermate controls. **d** Numbers of IL17 and IL22 producing ILC3 from adult Ccl19-Cre Ltbr[fl/fl] mice and co-housed littermate controls. **e** Relative expression of Reg3g, S100a8, and S100a9 from the small intestine of adult Ccl19-Cre Ltbr[fl/fl] mice and co-housed littermate controls measured by real-time PCR. **f** Bacterial concentration in faeces and colonic tissue from Ccl19-Cre Ltbr[fl/fl] mice and co-housed littermate controls on day 11 after C. rodentium infection. **g** Weight change from Ccl19-Cre Ltbr[fl/fl] mice and co-housed littermate controls on day 11 after C. rodentium infection. **h** Colon length from Ccl19-Cre Ltbr[fl/fl] mice and co-housed littermate controls on day 11 after C. rodentium infection. **i** Representative images of intestinal sections and the histological scores from Ccl19-Cre Ltbr[fl/fl] mice and co-housed littermate controls on day 11 after C. rodentium infection. **a–c** n = 11 and 12 mice from four independent experiments, mean ± SEM. **d** n = 6 and 9 mice from three independent experiments, mean ± SEM. **e** n = 10 mice per group from four independent experiments, mean ± SEM. **f** n = 8 and 12 mice from three independent experiments, geometric mean ± SD. **g–i** n = 8 and 12 mice from three independent experiments, mean ± SEM. Statistical analyses were performed using unpaired two-tailed Student's t test (**b–d**) and non-parametric two-tailed Mann–Whitney test (**e–g**, **i**).

could be detected in the SILTs (Fig. 4b). While mILF could still be detected after 2 weeks of Dox removal, mILFs were not detectable after prolonged cessation of Dox exposure (Dox off 8 wk) in Ccl19-iEYFP Ltbr[fl/fl] mice when compared to co-housed littermate controls (Fig. 4b, c). Quantitative image analysis confirmed that the maturation of SILT structures required continued LTβR signaling in Ccl19-expressing FRCs (Fig. 4c). In 16-week-old Ccl19-iEYFP Ltbr[fl/fl] mice, Dox removal for 8 weeks led to a significant reduction of ILC3 and ILC1 numbers in the lamina propria (Fig. 4d). Moreover, the infection of 16-week-old Ccl19-iEYFP Ltbr[fl/fl] mice (Dox off 8 wk) with C. rodentium revealed increased susceptibility of the mice when LTβR signaling was abrogated specifically in adult Ccl19-expressing FRCs (Fig. 4e). Moreover, we observed significant weight loss (Fig. 4f) and significantly reduced colon length (Fig. 4g) after C. rodentium infection under conditions of induced Ltbr-deficiency in adult SILT FRCs. These data indicate that SILT FRCs constantly interrogate their environment and receive crucial signals via the LTβR to nourish ILC niches and sustain SILT structures.

**Interactome analysis of FRC-ILC crosstalk.** To elaborate on the mechanisms underlying SILT FRC-mediated ILC sustenance, we analyzed the transcriptome of small intestinal CD127[+] ILCs from Ccl19-EYFP Ltbr[fl/fl] mice and co-housed littermate controls. The merged scRNA-seq datasets identified 8 main CD127[+] ILCs clusters that could be categorized based on particular gene expression pattern and known canonical markers[4,5,39] as Ncr[+] Rorc[+] ILC3, Ccr6[+] Rorc[+] ILC3, Gata3[+] Il1rl1[+] ILC2, Gata3[+] Il1rl1[-] ILC2 and Tbx21[+] ILC1 (Fig. 5a and Supplementary Fig. 7a). In addition, we identified two clusters that co-expressed the ILC2 marker Gata3 and the ILC3 marker Rorc, and showed elevated expression of key transcription factors found in ILC progenitors (ILCp) such as Nfil3[40,41], Tcf7[42], and Tox[43] (Fig. 5a, b). These data suggest that cells in the ILCp cluster possess the potentials for mixed-lineage contribution[44]. One of the two ILCp clusters expressed ILC2 activation markers such as Ly6a and Klrg1 suggesting a poised ILC2 lineage differentiation[45] (Supplementary Fig. 7a). Hence, the ILCp clusters were labeled as ILC2p and ILCp, respectively (Fig. 5a and Supplementary Fig. 7a). The extended analysis of the combined scRNA-seq datasets from Ccl19-EYFP Ltbr[fl/fl] mice and the littermate controls revealed

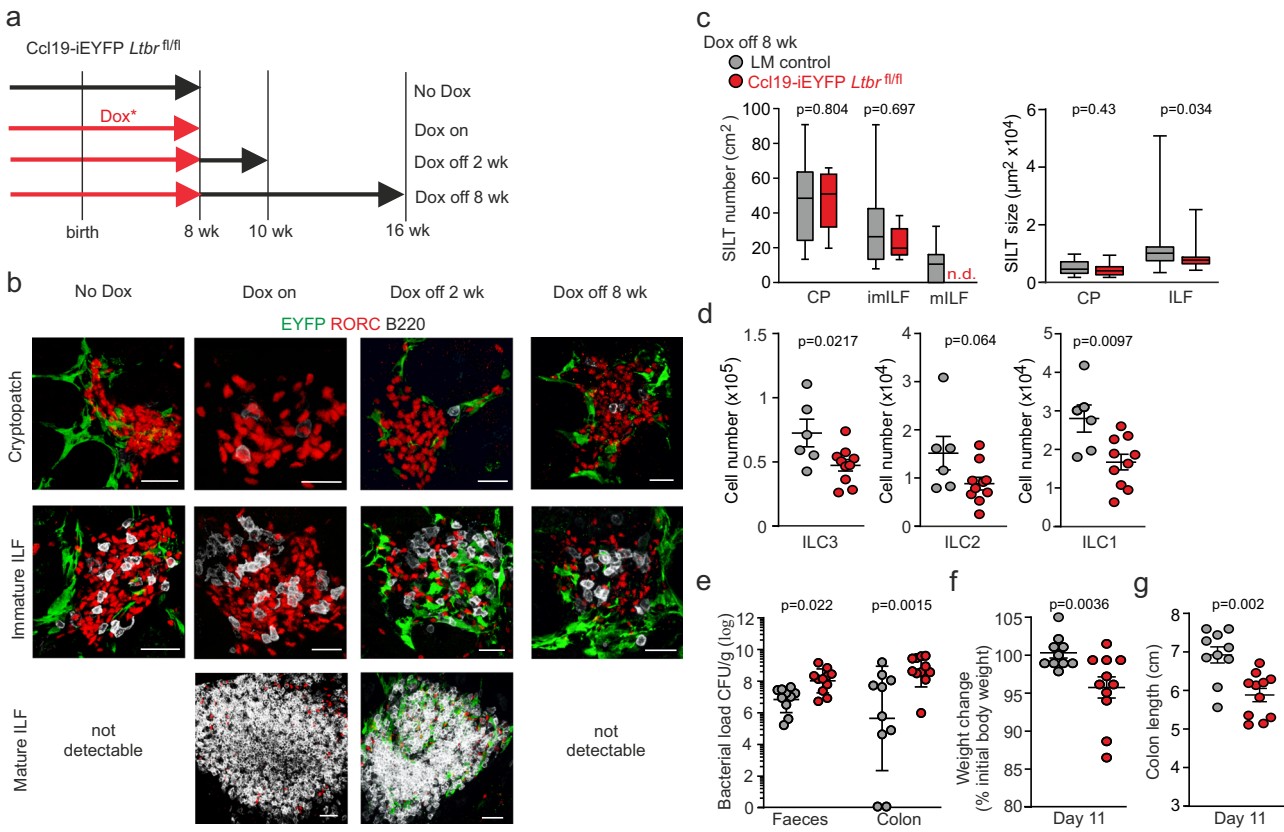

**Fig. 4 SILT maturation and ILC sustenance in the absence of persistent LTβR signaling in FRCs. a** Schematic timeline of non-antibiotic Dox treatment regime **b** Representative images of SILT structures in different conditions analyzed by confocal microscopy after staining with the indicated antibodies. Scale bar, 20 μm in CP, 30 μm in imILF, and 30 μm in mILF. **c** Number and size of SILT structures detected in the small intestine from Ccl19-iEYFP *Ltbr*^fl/fl mice and co-housed littermate controls after Dox withdrawal for 8 weeks. **d** Cell number of ILC subsets from Ccl19-iEYFP *Ltbr*^fl/fl mice and co-housed littermate controls after Dox withdrawal for 8 weeks. **e** Bacterial concentration in faeces and colonic tissue on day 11 after *C. rodentium* infection of Ccl19-iEYFP *Ltbr*^fl/fl mice and co-housed littermate controls after Dox withdrawal for 8 weeks. **f, g**. Weight change (**f**) and colon length (**g**) on day 11 after *C. rodentium* infection in Ccl19-iEYFP *Ltbr*^fl/fl mice and co-housed littermate controls after Dox withdrawal for 8 weeks. **b** Images are representative of at least four mice. **c** $n = 13$ and 8 mice from three independent experiments in the left panel, $n = 27$ and 18 in CP and $n = 26$ and 16 in ILF from 5 mice with two independent experiments in the right panel. median ± interquartile ranges. **d** $n = 6$ and 10 mice from three independent experiments, mean ± SEM. **e** $n = 10$ and 11 mice from three independent experiments, geometric mean ± SD. **f, g** $n = 10$ and 11 mice from three independent experiments, mean ± SEM. Statistical analyses were performed using non-parametric two-tailed Mann–Whitney test (**c, e–g**) and unpaired two-tailed Student's *t* test (**d**).

comparable ILC subset composition (Fig. 5a), a result that confirmed our flow cytometry analysis shown in Fig. 3a.

To assess the full array of molecular interactions between SILT FRCs and ILCs, we used the CellPhone-DB algorithm[46]. In this analysis, pairs of known interacting ligands and receptors are derived from public databases and cell types are analyzed for enriched receptor–ligand interactions based on the expression of a receptor by one cell type and a ligand by another cell type[46]. Using this approach, we found that the number of detected ligand and receptor interactions between FRCs and all ILC subsets except for ILC2p were enriched under *Ltbr*-proficient conditions (Fig. 5c) suggesting that distinct molecular circuits in SILT FRC niches drive the interaction with ILCs. The subsequent in silico interactome analysis indicated the major predicted SILT FRC niche factors (IL7, IL15, Dll1, KITL), chemokines (CXCL13 and CXCL12), and adhesion molecules (VCAM1) that direct interaction with the different ILC subsets (Fig. 5d and Supplementary Fig. 7b). Moreover, the algorithm identified ILC-derived ligands such as lymphotoxin-α (LTA) that contribute to the reciprocal crosstalk between the cell populations (Fig. 5d). In accordance with our experimental observations of *Il7* downregulation in the absence of LTβR signaling (Fig. 2e, f), the predicted interactions in the IL7-IL7R axis were substantially reduced in some or all ILC

subsets (Fig. 5d). Since IL7 availability was shown to be important for sustained ILC programming[47,48], we hypothesized that IL7 provided by SILT FRCs may act as one of the key niche factors for ILC homeostasis. We therefore crossed Ccl19-EYFP mice with *Il7*^fl/fl mice and found that conditional ablation of *Il7* in FRC did not affect ILF maturation (Supplementary Fig. 7c), but significantly reduced accumulation of RORC+ ILCs in imILFs and mILFs (Fig. 5e and Supplementary Fig. 7c). Flow cytometric analysis of ILC populations revealed a substantial reduction in total ILC numbers and in all ILC subsets in the lamina propria of Ccl19-Cre *Il7*^fl/fl mice in comparison to co-housed Cre-negative littermate controls (Fig. 5f). Consistent with the data from mice with *Ltbr*-deficient SILT FRC niches, the overall ILC composition remained comparable under *Il7*-proficient and -deficient conditions (Supplementary Fig. 7d). However, conditional ablation of *Il7* gene in SILT FRCs did not affect T cell subset composition in the lamina propria of Ccl19-Cre *Il7*^fl/fl mice when compared to co-housed littermate controls (Supplementary Fig. 7e–g). The infection of Ccl19-Cre *Il7*^fl/fl mice with *C. rodentium* further showed shortened colon length (Supplementary Fig. 7h) and increased bacterial concentration in the faeces and colon (Supplementary Fig. 7i). Taken together, the FRC-ILC interactome analysis unveiled critical niche factors and molecular

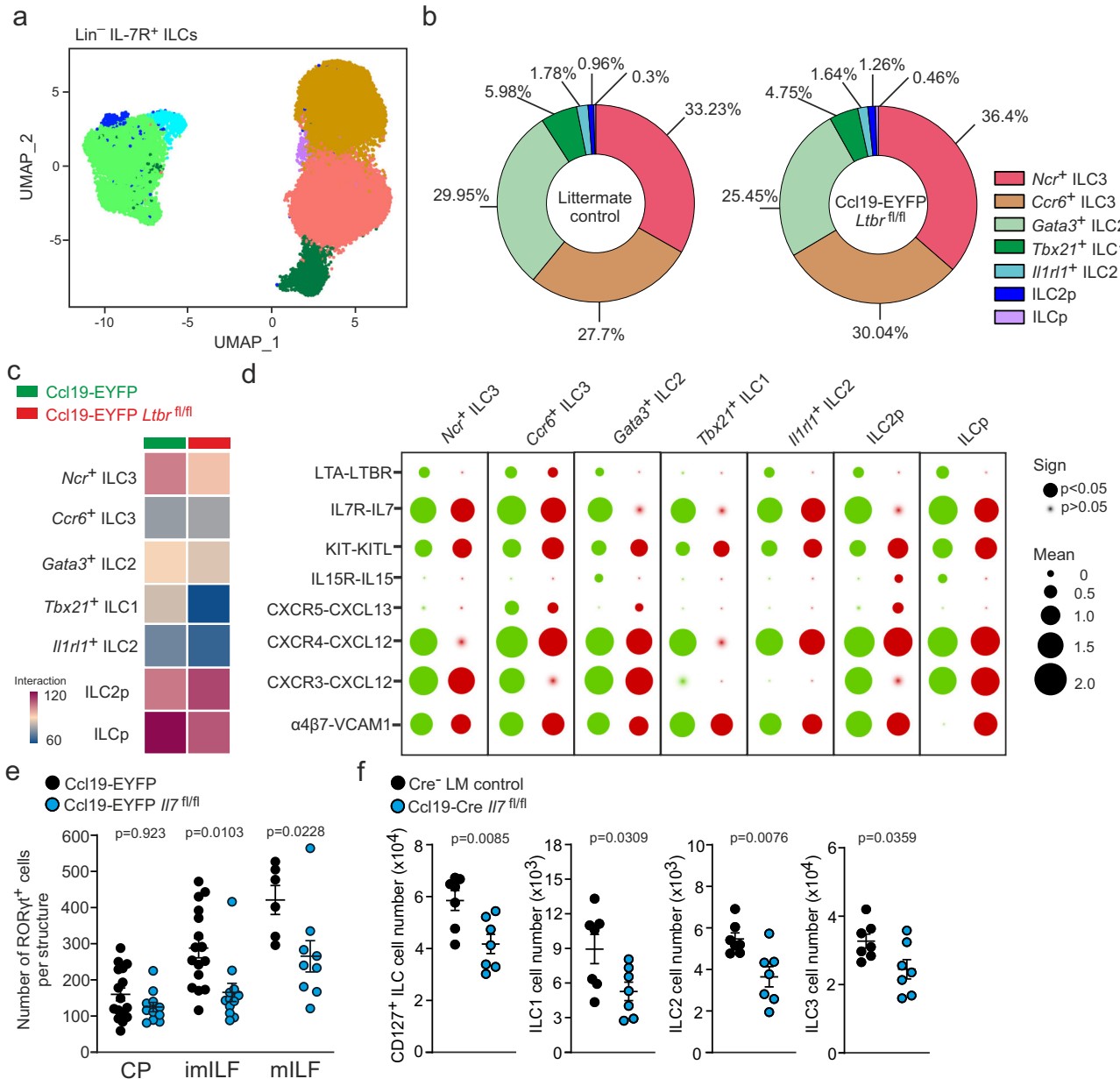

**Fig. 5 Interactome analysis of SILT FRC and lamina propria ILCs. a** UMAP of sorted CD127+ ILCs from Ccl19-EYFP *Ltbr*fl/fl and co-housed littermate controls. **b** ILC composition based on scRNA-seq data. **c** Heatmap indicating the number of ligand and receptor interactions between SILT FRC and ILCs from Ccl19-EYFP and Ccl19-EYFP *Ltbr*fl/fl mice using CellPhone analysis. **d** Predicted intensity of ligand and receptor interaction between SILT FRCs and ILCs from Ccl19-EYFP and Ccl19-EYFP *Ltbr*fl/fl mice based on the CellPhone analysis. The opacity of the circles indicates the degree of statistical significance and the size of the circles provide a measure of the interaction intensity. **e** Number of RORC+ ILC3 residing in SILTs of Ccl19-EYFP and Ccl19-EYFP *Il7*fl/fl mice. **f** Number of CD127+ ILCs and ILC subsets from adult Ccl19-Cre *Il7*fl/fl mice and co-housed littermate controls. **a** ScRNA-seq data represents 40,473 CD127+ ILCs, n = 6 biological replicates from Ccl19-EYFP *Ltbr*fl/fl mice and co-housed littermate controls. **e** n = 16 and 11 in CP, n = 16 and 12 in imILF, n = 6 and 9 in mILF of 3 mice from two independent experiments in each condition, mean ± SEM. **f** n = 7 and 7 mice from three independent experiments, mean ± SEM. Statistical analyses were performed using one-way ANOVA with Tukey's multiple comparison test (**e**) and unpaired two-tailed Student's *t* test (**f**).

interaction partners that support ILC sustenance and function in SILTs.

## Discussion

The intestinal immune system has to react to the constantly evolving microbiome with varying composition of commensals, changes in pathobiont communities, and occasional exposure to pathogens. While the number and location of Peyer's patches in the small intestine are programmed during embryogensis[35], the

postnatal development of SILTs provides the flexibility to establish additional lymphoid organs on demand. The data presented here show that number, size, structure, and function of the inducible lymphoid organs in the small intestine depends on the interaction between *Ccl19*-expressing SILT FRCs and ILCs. The combination of inducible gene ablation and transcriptomics-based interactome analysis revealed the tonic nature of the SILT FRC-ILC crosstalk. SILT FRCs integrate signals from ILCs to generate and maintain the niche environment required for ILC sustenance. Disruption of the core

signaling pathways underlying FRC-ILC interaction led to the dismantling of SILT structures and impaired ILC-mediated immune protection. Thus, a small (<2%) fraction of highly specialized fibroblasts in the intestinal lamina propria controls ILC homeostasis and function.

Signaling via the LTβR is the key process underlying the initiation of SLO formation during embryogenesis[19] and post-natal generation of SILTs[22]. However, a large array of immune cells and non-hematopoietic cells exhibit *Ltbr* expression[49] and recent studies have shown that LTβR-positive stromal cell populations contribute to different degrees to lymphoid organ development[35,50,51]. Our findings support the notion that SILT formation and maturation is regulated to a large extent by LTβR-signaling in Ccl19-Cre+ FRCs and their progenitors. This scenario is highly likely because LTβR-signaling in neither intestinal epithelial cells nor dendritic cells contributes to SILT formation[52,53]. Thus, SILT development and the steering of immune cell interactions in SILTs are controlled by a sequence of events that starts with an enabling "signal 1" in Ccl19-Cre-expressing FRCs, i.e., LTβR-signaling, and a series of further molecular FRC-ILC interactions that facilitate the full functionality of these lymphoid tissues.

A recent scRNA-seq-based analysis of fibroblastic stromal cells from patients with inflammatory bowel disease has revealed the presence of a *CCL19*-expressing cell population in the inflamed tissues[54]. Cells in this particular cluster express the SILT FRC signature genes *CCL21* and *CLU*[54]. It is still unclear whether *CCL19*-expressing cells in the human lamina propria are located exclusively in the SILTs as demonstrated here for the mouse small intestine. Nevertheless, the increase of organized lymphoid structures in numbers and size in Crohn's disease patients[55,56] suggests that chronic intestinal inflammation is associated with SILT formation in humans. Furthermore, aberrant ILC composition[57] and increased IL23-mediated IL17A production by ILCs[58] were shown to be associated with Crohn's disease supporting the notion that ILC activity in the human intestine might be controlled in organized lymphoid structures. It is noteworthy that Ccl19-Cre+ FRCs do not only underpin the fibroblastic infrastructure of SLOs[32,34,35,59], but participate in the formation of inducible tertiary lymphoid tissues such as bronchus-associated lymphoid tissues in the lung during chronic immune activation[60]. It will be important to elaborate in future studies on whether and to what extent the excessive formation of SILTs and/or tertiary lymphoid tissues in the intestinal lamina propria can be modulated to attenuate inflammatory reactions in human disease. Clearly, the data presented here show that an impairment of the ILC niche formed by SILT FRCs bears the danger of reducing the ability of the intestinal immune system to cope with pathogens.

The bone marrow niche concept delineates the micro-environmental conditions that facilitate the maintenance and differentiation of hematopoietic stem cells[61]. Recent amendments of the concept with consideration of peripheral immune processes highlight the general relevance of niche conditions for immune cells such as macrophages and ILCs: provision of scaffold, supply of trophic factors, induction of tissue imprinting, and reciprocal benefits during cell-cell interactions[62]. SILT FRCs fulfill all criteria of a critical niche-forming cell. The dense cellular FRC network generates a distinct space for immune cell interactions and provides ample surface for direct FRC-ILC communication. The interactome analysis based on single-cell transcriptomics data from both FRCs and ILCs revealed the major pathways involved in the communication between the two cell populations. LTβR-dependent ILC stimulation by FRCs includes the production of IL7 to sustain the size of the ILC pools. In addition, the interactome analysis suggests that several LTβR-independent reciprocal interactions are operational, e.g., IL33-IL1RL1, KITL-

KIT and CXCL16-CXCR6, to mold the niche microenvironment. It remains to be determined which cell type provides the stimuli that initiate FRC-ILC interaction in CPs prior to LTβR-dependent cascade. It is possible that glial cells, which protrude into the SILT structures and can be found in close contact with ILC3[63], generate neurotrophic factors that contribute to these processes. Further elaboration of the initiation of FRC-ILC interaction in SILTs and the means of their reciprocal interaction will be important to better understand the basis of intestinal immunity.

In sum, our study disentangles core processes that highlight immune cell interactions in the intestinal lamina propria and control of pathogens. A distinct fraction of highly specialized fibroblasts appears to be at the nexus of immune cell interactions in the lamina propria and thereby controls the balance between immunity and immunopathology. Hence, the identification of drugable targets in SILT FRCs can guide the development of therapeutic approaches to attenuate excessive inflammatory processes in the gut.

## Methods

**Mice**. All mouse strains were on a C57BL/6NCrl genetic background and maintained in individually ventilated cages under specific pathogen-free conditions. C57BL/6N-Tg(Ccl19-Cre)489Biat (*Ccl19-Cre*)[34], C57BL/6N-Tg(Ccl19-tTA)688BIAT (*Ccl19-tTA*)[32], and Ltbrtm1.1Thhe (*LTβR*fl/fl)[64] strains were described previously. C57BL/6N-(IL7fl/fl)tm1Iku (*IL7*fl/fl)[65] are kindly provided by Koichi Ikuta (Kyoto University, Japan). B6.129 × 1-Gt(ROSA)26Sortm1(EYFPCos)/J (R26R-EYFP) mice were purchased from The Jackson Laboratories and the LC1 strain[66] was kindly provided by Dr Fendler (Max Delbrück in Center of Molecular Medicine, Berlin, Germany). Non-antibiotic 4-Epidoxycycline (Dox) was kindly provided by Dr Rodewald at the University of Leipzig, Germany. 4-Epidoxycycline was administered to pregnant dams in the drinking water (50 μg/ml) and maintained after weaning. All mice were housed in the Institute of Immunobiology, Kantonsspital St. Gallen under specific pathogen-free conditions at 22 ± 2 °C and 50 ± 5% humidity in a 12/12 h light/dark cycle and provided ad libitum access to food and water. Experiments were performed in accordance with Swiss federal and cantonal guidelines (Tierschutzgesetz) under the permission numbers SG01/18, SG08/18, SG04/20, and SG24/2020 granted by the Veterinary Office of the Canton of St. Gallen.

**Antibodies**. Anti-Mouse CD3ε Biotin (Clone:145-2C11, 100303, 1:100), Anti-Mouse GR1 Biotin (Clone:RB6-8C5, 108403, 1:100), Anti-Mouse CD19 Biotin (Clone: 6D5, 115503, 1:100), Anti-Mouse Ter119 Biotin (Clone: Ter119, 116203, 1:100), Anti-Mouse CD5 Biotin (Clone:53-7.3, 100603, 1:100), Anti-Mouse CD127/ IL7R BV711 (Clone:A7R34, 135035, 1:50), Anti-Mouse CD45.2 BV605 (Clone:104, 109841, 1:100), Anti-Mouse CD335/Nkp46 BV421 (Clone:29A1.4, 137612, 1:100), Anti-Mouse T-bet PE-Cy7 (Clone:4B10, 644823, 1:100), Anti-Mouse CD8 BV711 (Clone:53-6.7, 100747, 1:100), Anti-Mouse CD4 BV605 (Clone:RM4-5, 100547, 1:100), Anti-Mouse CD45.2 BV421 (Clone:104, 109831, 1:100), Anti-Mouse CD3ε PerCP (Clone:145-2C11, 100325, 1:100), Anti-Mouse/Human GL-7 AlexaFluor488 (Clone:GL-7, 144611, 1:100), Anti-Mouse CD11c APC-Cy7 (Clone:N418, 117323, 1:100), Anti-Mouse I-A/I-E BV421 (Clone:M5/114.15.2, 107631, 1:100), Anti-Mouse/Human CD11b AlexaFluor488 (Clone:M1/70, 101219, 1:100), Anti-MouseLy6G PE (Clone:1A8, 127607, 1:100), Anti-Mouse F4/80 PE-Cy7 (Clone:BM8, 123113, 1:100), Anti-Mouse/Human Ki67 AlexaFluor488 (Clone:11F6, 151204, 1:200), Anti-Mouse IL17A AlexaFluor488 (Clone:TC11-18H10.1, 506909, 1:50), Anti-Mouse CD81 PerCP-Cy5.5 (Clone:Eat-2, 104911, 1:100), Anti-Mouse PDPN APC-Cy7 (Clone:8.1.1, 127417, 1:200), Anti-Mouse CD90.2 BV785 (Clone:30-H12, 105331, 1:100), Anti-Mouse TER-119/Erythroid Cells BV605 (Clone:TER-119, 116239, 1:100), Anti-Mouse CD34 BV421 (Clone:MEC14.7, 119321, 1:100), Anti-Mouse CD31 PE-Cy7 (Clone:390, 102418, 1:100), Anti-Mouse CD326 (EpCAM) PE-Cy7 (Clone:G8.8, 118216, 1:100), Anti-Mouse/Human CD45R/B220 AlexaFluor647 (Clone:RA3-6B2, 103226, 1:500), Syrian Hamster Anti-Mouse Podoplanin (Clone:8.1.1, 127401, 1:500), Anti-Mouse CD31 Alexa-Fluor647 (Clone:MEC13.3, 102515, 1:500), Biotin Anti-Mouse CD90.2 (Thy1.2) (Clone:53-2.1, 140314, 1:500), Anti-Mouse TCR γ/δ APC (Clone:GL3, 118115, 1:100) were purchased from BioLegend. Fixable Viability Stain 510 (Aqua) (564406, 1:1000), Anti-Mouse CD196(CCR6) AlexaFluor647 (Clone:140706, 557976, 1:50), Anti-Mouse RORγt PerCP-Cy5.5 (Clone:Q31-378, 562683, 1:50), Anti-Mouse Ly6C PerCP-Cy5.5 (Clone:AL-21, 560525, 1:100), Anti-Mouse CD56/ NCAM1 BV711 (Clone:809220, 565987, 1:100) and Anti-Mouse CD54 BV421 (Clone:3E2, 565987, 1:100) were purchased from BD Bioscience. Fixable viability dye eFluor780 (Live/Dead) (65-0865-14, 1:1000), Rat Anti-Mouse ROR gamma t (Clone:AFKJS-9, 14-6988-82, 1:50), Anti-Mouse GATA-3 PE (Clone:TWAJ, 12-9966-42, 1:100) Anti-Mouse EOMES AlexaFluor488 (Clone:Dan11mag,

53-4875-82, 1:100), Anti-Mouse IgA PE(Clone:11-44-2, 12-5994-81, 1:20), Anti-Mouse IL22 PE (Clone:1H8PWSR, 12-7221-82, 1:20), Anti-Mouse/Human Alpha-Smooth Muscle Actin eFluor660 (Clone:1A4, 50-9760-82, 1:100), Anti-Mouse FOXP3 PE-Cy7 (Clone: FJK-16s, 891552, 1:100) and Anti-Mouse B220/CD45R APC-Cy7 (Clone:RA3-6B2, 552094, 1:200) were purchased from eBioscience. Anti-Mouse/Human Alpha-Smooth Muscle Actin Cy3 (Clone:1A4, C6198, 1:500) was purchased from Merck, Goat Anti-Mouse Clusterin (Clone:AF2747, 1:200), Goat Anti-Mouse/Human NCAM1/CD56 (Clone:AF2408, 1:500), Biotin Anti-Mouse CXCL13/BLC/BCA-1 (Clone:BAF470, 1:100), Biotin Anti-Human/Mouse Tenas-cinC (Clone:MAB2138, 1:500) were purchased from R&D system. Rabbit Anti-Full-Length GFP Polyclonal Antibody (Clone: 632592, 1:500) was purchased from Takara Bio Clontech. Anti-Mouse SiglecF APC (Clone:ES22-10D8, 130-123-816, 1:100) was purchased from Miltenyi Biotec. CD1d/PBS-57-Tetramer PE (1:50) was kindly provided by Dr Stefan Freigang from University of Bern. Unconjugated antibodies were stained following with secondary antibodies: Alexa488-conjugated anti-rabbit-IgG (711-546-152, 1:500), Alexa488-conjugated anti-rat-IgG (112-545-003, 1:500), Dylight549-conjugated anti-syrian hamster-IgG (107-505-142, 1:500), Cy3-conjugated anti-Biotin (200-162-211, 1:500), Alexa647-conjugated anti-Biotin (200-602-211, 1:500), Alexa647-conjugated anti-goat-IgG (705-605-003, 1:500), Cy3-conjugated anti-goat-IgG (705-605-003), Biotin-conjugated anti-rat-IgG (112-065-003, 1:500) purchased from Jackson Immunotools.

**Isolation of distinct cell populations from the small intestine.** For the isolation of fibroblasts from the lamina propria, tissue was harvested and incubated three times for 15 min at room temperature under constant agitation with PBS containing 5% FCS (Sigma-Aldrich, F2442), 5 mM EDTA (Sigma-Aldrich, E9884), 10 mM HEPES (Sigma-Aldrich, H0887) and 1 mM DTT (Applichem, 3483-12-3) in order to dissociate the epithelial layer. The tissue was subsequently washed with BSS containing 10 mM HEPES and digested three times for 20 min at 37 °C under constant agitation with 120 μg/ml collagenase P (Roche, 11215809103), 25 μg/ml DNase I (Applichem, A37780010) and 5 μg/ml Dispase I (Roche, 4942078001) in medium (RPMI-1640 (Sigma-Aldrich, R8758) containing 5% FCS (Sigma-Aldrich) and 10 mM HEPES (Sigma-Aldrich)). To enrich lamina propria fibroblasts, hematopoietic and erythrocytes were depleted by incubating the cell suspension with MACS anti-CD45 (Miltenyi Biotec, 130-052-301) and anti-Ter119 microbeads (Miltenyi Biotec, 130-049-901) and passing through a MACS LS column (Miltenyi Biotec, 130-042-401). Unbound single-cell suspensions were stained with different markers for further flow cytometric analysis.

For the isolation of lymphocytes and ILCs from the lamina propria, the protocol was followed as above-described for fibroblasts, but the enzymatic digestion mixture contained only 120 μg/ml collagenase P (Roche, 11215809103) and 25 μg/ml DNase I (Applichem, A37780010). Instead of treating with microbeads, single-cell suspension was resuspended with 5 ml 30% Percoll diluted with RPMI-1640 medium (Sigma-Aldrich) containing 5% FCS (Sigma-Aldrich) and 10 mM HEPES (Sigma-Aldrich)) and was underlayed with 3 ml 70% Percoll (Cytiva, 17089101) diluted with RPMI-1640 medium containing 5% FCS and 10 mM HEPES, and centrifuged at 500 g for 20 min. After centrifugation, the lymphocyte/ILC layer was retrieved and washed with MACS buffer. This cell fraction was stained and processed for flow cytometric analysis.

**Antibody staining for sorting and flow cytometric analysis.** Cell suspensions of lamina propria fibroblasts were prepared as described above and incubated with the Live/Dead Fixable Aqua staining, and stained with different fluorochrome-conjugated antibodies against the following surface markers: PDPN, CD81, Thy1.2, NCAM1, CD45/Ter119, CD34, CD31, and EpCAM. Next, intracellular staining was performed through the treatment with FoxP3/transcription factor staining buffer (eBioscience, 00-5521-00) for 30 min and stained with antibodies against ACTA2 and Clusterin in permeablization buffer as indicated in the protocol provided by the manufacturer.

Cell suspensions of hematopoietic cells were prepared as described above and incubated with the Live/Dead Fixable Aqua staining, and stained with fluorochrome-conjugated antibodies. For ILC staining, cell fractions were stained with antibodies against CD127, CD45.2, Nkp46, CCR6 together with biotin-labeled anitbodies against lineage (Lin) markers (CD19, CD3, GR1, Ter119, and CD5), followed by APC-Cy7-conjugated streptavidin staining. Next, the cell fraction was treated with FoxP3/transcription factor staining buffer (eBioscience, 00-5521-00) and subsequently stained for transcription factors GATA3, RORC, Tbet, and Eomes or cell proliferation marker Ki67[67]. For the apoptosis assay, the CellEvent Caspase 3/7 green flow cytometry assay kit (ThermoFisher, C10740) was utilized and performed according to the instructions of the manufacturer.

For IL17 and IL22 staining, the ILC cell fraction was stimulated with 10 ng/ml IL1β (R&D system, 401-ML-025), 25 ng/ml IL23 (R&D system, 1887-ML-010) and 10 μg/ml Brefeldin A (Sigma-Aldrich, B6542) in RPMI medium with 1 mM HEPES, 1% sodium pyruvate (Gibco, 11360070), 10% FCS and 1% NEAA (Gibco, 11140050) for 4 h at 37 °C. After stimulation, the cell fraction was centrifuged and was first stained with the antibodies against CD127, CD45.2, Nkp46, CCR6 together with lineage (Lin) markers, followed by APC-Cy7-conjugated streptavidin staining. Next, the cell fraction was treated with FoxP3/transcription factor staining buffer for 12 h and subsequently stained for RORC, Tbet, IL17, and IL22.

For lymphocytes staining, cell suspensions were incubated with fluorochrome-conjugated antibodies against CD45.2, CD3, CD4, CD8, B220, and GL-7. This cell fraction was further treated with FoxP3/transcription factor staining buffer and subsequently stained for IgA or FoxP3. For Th17 staining, cell suspensions were stimulated with PMA/ionomycin (Sigma-Aldrich, P8139/I0634) and 10 μg/ml Brefeldin A in the RPMI medium with 1 mM HEPES, 1% sodium pyruvate, 10% FCS and 1% NEAA for 4 h at 37 °C. After stimulation, the cell fraction was centrifuged and was first stained with the antibodies CD45.2, CD3, CD4, CD8, and γδ TCR for 30 min at 4 °C. This cell fraction was further treated with FoxP3/transcription factor staining buffer and subsequently stained for IL17. For myeloid cell staining, cell suspensions were incubated with fluorochrome-conjugated antibodies against CD45.2, CD11c, MHCII, SiglecF, Ly6C, Ly6G, and F4/80. Cells were washed either with MACS buffer or with permeabilization buffer, acquired with a BD LSR Fortessa (BD Biosciences) in software FACSDiva (BD Biosciences), and analyzed using FlowJo software v10.1r5 (Tree Star Inc.).

**Histology.** Organs were fixed overnight with 4% paraformaldehyde (Merck-Millipore, 104005) in PBS under agitation at 4 °C. Fixed samples were further washed with PBS containing 1% TritonX-100 (Sigma-Aldrich, T8787) and 2% FCS (Sigma-Aldrich) overnight at 4 °C. Samples were embedded in 4% Nusieve GTG low melting agarose gel (Lonza Bioscience, 50080) and 40 μm sections were generated with a vibrotome (Leica VT-1200). Sections were blocked in PBS containing 1% TritonX-100 (Sigma-Aldrich), 10% fetal calf serum (Sigma-Aldrich) and 1 mg/ml anti-Fcγ receptor (BD Biosciences, AB_394656) at 4 °C for 2 h and further incubated overnight with the following labeled antibodies: anti-PDPN (eBioscience), anti-B220 (eBioscience), anti-ACTA2 (Sigma-Aldrich), anti-EYFP (Clontech), anti-Clusterin (R&D), anti-NCAM1 (R&D), anti-CXCL13 (R&D), anti-CD31 (BioLegend), anti-THY1 (BioLegend) and anti-TNC (R&D). Unconjugated antibodies were detected with the following secondary antibodies: Alexa488-conjugated anti-rabbit-IgG, Dylight549-conjugated anti-rat-IgG, Dylight549-conjugated anti-syrian hamster-IgG, Cy3-conjugated Streptavidin, Alexa647-conjugated Streptavidin, Alexa647-conjugated anti-goat-IgG (all purchased from Jackson Immunotools). For RORC staining, intestinal samples were processed with FoxP3/transcription factor staining buffer set (eBioscience). Briefly, samples were incubated with FoxP3/transcription factor staining buffer overnight. After washing with permeabilization buffer, samples were stained with rat anti-RORC (eBioscience) antibody for 6 h at 4 °C. Samples were then stained with biotin-conjugated anti-rat secondary antibody overnight after washing with permeabilization buffer. Finally, samples were incubated with Cy3-conjugated streptavidin and anti-B220 (eBioscience) for 2 h. Microscopy analysis was performed using a confocal microscope (Zeiss LSM-710) and images were processed with ZEN 2010 software (Carl Zeiss, Inc.) and Imaris (Bitplane).

**Quantification of CP and ILF structures.** Samples from the distal ileum were fixed, cut, and stained as described above. For the quantification of CPs and ILFs, the DAPI area of three to six sections per mouse was recorded. CPs were defined as clusters of EYFP+ fibroblastic stromal cells, but containing less than 10 B220+ B cells. ImILFs were defined as clusters of RORC+ ILC3s with more than 10 B cells. mILFs were defined as cell accumulations with a large central B-cell cluster and a corona of RORC+ ILC3s. The sum of counted CPs and ILFs was normalized to an area of 1 cm² DAPI area. The area covered by CPs and ILFs was calculated on the basis of RORC+ signal within CPs and ILFs using the ZEN 2010 software (Carl Zeiss Inc.). Quantification of RORC+ ILC3 numbers was evaluated through the colocalization of DAPI and RORC signals within defined SILT structures reconstructed in 3D in Imaris (Bitplane) using an automatic threshold.

**Infection of *Citrobacter rodentium*.** *C. rodentium* cultures (strain ICC169, kindly provided by K. Maloy, Oxford, UK) were grown from a single colony overnight in LB broth supplemented with 50 μg/ml nalidixic acid (Sigma-Aldrich). Mice were fed ~5 × 10⁹ bacteria by oral gavage in 200 μl PBS. For the analysis of bacterial burdens, mice were sacrificed and samples were taken 11 days post infection from caecum, colon, and feces. Colon length was measured and the collected samples were mechanically homogenized and cultured overnight on LB agar plates supplemented with 50 μg/ml nalidixic acid. Samples from the colon were fixed in 4% formalin and embedded in paraffin. Samples were cut into 1.5-μm sections and stained with hematoxylin and eosin. To determine the histopathological score, the following parameters were considered and graded by two blinded examiners on a scale from 0 (none) to 3 (severe) for a maximal histology score of 12: (1) cellular infiltration, (2) erosion, (3) crypt architectural distortion, and (4) edema.

**Quantitative real-time PCR.** Total cellular RNA was extracted from the sorted cells using Quick-RNA microprep kit (Zymo Research, R1050) or from tissue using Direct-zol RNA Miniprep kit (Zymo Research, R2051) following the commercial protocol. cDNA was prepared using High Capacity cDNA Reverse Transcription kit (Applied Biosystems, 4368814) and quantitative PCR was performed using PoweUp SYBR Green master mix (Applied Biosystems, A25741) on a QuantStudio 5 machine (Applied Biosystems) and analyzed with the QuantStudio™ Design & Analysis Software (Applied Biosystems). Expression levels were measured by using

the following primers: Il7 (QT00101318, Qiagen), Ccl19 (QT02532173, Qiagen), Clu (QT00146174, Qiagen), Il33 (QT00135170, Qiagen). RegIIIg (fw: ATGGCTCCTATTGCTATGCC, rv: GATGTCCTGAGGGCCTCTT), S100a8 (fw: TGTCCTCAGTTTGTGCAGAATATAAA, rv: TCACCATCGCAAGGAACTCC), S100a9 (fw: GGTGGAAGCACAGTTGGCA, rv: GTGTCCAGGTCCTCCAT-GATG) were synthesized by Microsynth.

**Droplet-based single-cell RNA-seq analysis.** Lamina propria fibroblasts and EYFP+ cells from the small intestine of Ccl19-EYFP and Ccl19-EYFP Ltbr^fl/fl mice were sorted with a BD FACS Melody cell sorter (BD Biosciences) and analyzed with FACSChorus (BD Biosciences). For the scRNA-seq analysis of ILCs, the cell suspension from the small intestine of Ccl19-EYFP Ltbr^fl/fl mice and the co-housed littermates were enriched for ILCs via the Easysep Mouse Pan ILC enrichment kit (StemCell Technology) and staining with the lineage cell markers and CD127 as described above for sorting. Sorted single-cell suspensions were run on the 10x Chromium analyzer (10X Genomics)[68]. cDNA library generation was performed following the established commercial protocol for Chromium Single Cell 3' Reagent Kit (v3 Chemistry). Libraries were run via Novaseq6000 for Illumina sequencing. A total of 12 samples were processed in two batches, with all conditions present in both batches. Preprocessing and gene expression estimation were performed using CellRanger (v3.0.2)[69] with Ensembl GRCm38.9 release as a reference to build the index. Additional quality control and further analysis were run in R v.3.6.2 using the scater R/Bioconductor package (v1.14.6)[70] for quality control and the Seurat package (v.3.1.4)[71] for downstream analysis. First, in each sample cells with an exceedingly high or low number of detected genes or UMI counts (>2.5 median absolute deviations from the overall median) and cells with a large fraction of mitochondrial reads (>2.5 median absolute deviations above the median fraction) were excluded. In addition, known cell type markers were used to identify and remove contaminating or cycling cells from further analysis. More specifically, samples sorted for lamina propria fibroblasts were filtered for cells with no expression of Lyve1, Pecam1, Ptprc, Cd79a, L1cam, Mki67, and Sox10, and samples sorted for ILCs were filtered to remove cells expressing one of the markers Cd79a, Cd3e, Col3a1, or Itgam and only cells expressing Il7r were considered as ILCs and kept for further analysis. As a result, a total of 37,415 cells assigned to lamina propia fibroblasts (wild type: 25,614 cells, Ltbr^fl/fl: 11,801 cells) and 40,473 cells from samples sorted for ILCs (wildtype: 21,260 cells, Ltbr^fl/fl: 19,213 cells) were included for downstream analysis. Following quality control and filtering samples were merged in one dataset of lamina propia fibroblasts and one dataset of ILCs and both datasets were analyzed in parallel. Downstream analysis included normalization, scaling, dimensional reduction with PCA and UMAP, graph-based clustering, and calculation of unbiased cluster markers. Next, clusters were characterized based on the expression of calculated cluster markers and canonical marker genes reported in the previous publications[31–33]. Finally, cells from wild-type and Ltbr^fl/fl samples were compared by computing overall and clusterwise differentially expressed genes using the Wilcoxon test as implemented in the FindMarkers function of the Seurat package (v.3.1.4)[71] and visualized utilizing the pheatmap R package (v.1.0.12).

**CellPhone-DB analysis.** In order to analyze the interactome between SILT FRCs and ILCs, CellPhone-DB[46] was used as a tool to predict ligand–receptor interactions based on the processed scRNA-seq data. CellPhone-DB utilizes known ligand–receptor pairs from public databases to predict enriched receptor–ligand interactions between cell types. In brief, for each receptor–ligand pair, a null distribution is determined based on the average expression in randomly generated clusters derived from random permutations of cell type labels. Actual mean expression values from cell type pairs are then compared against null distributions to estimate the likelihood of cell-type specificity of each receptor–ligand pair. Here, SILT FRCs from wildtype and Ltbr^fl/fl samples were analyzed individually for enriched interactions with each of the ILC subsets. In order to ensure comparability between subsets and conditions all subsets for both conditions were downsampled to 200 cells before running CellPhone-DB (v.2.1.7)[46] with python v.3.7.0 and default parameters.

**Statistics.** GraphPad Prism 7 was used for all statistical analyses. Differences with a p value <0.05 were considered statistically significant.

**Reporting summary.** Further information on research design is available in the Nature Research Reporting Summary linked to this article.

## Data availability

The scRNA-seq data generated in this study have been deposited in the arrayexpress database (www.ebi.ac.uk/arrayexpress) under accession code E-MTAB-10638 and E-MTAB-10645. Source Data are provided with this paper.

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

## Acknowledgements

We thank Dr Stefan Freigang (Institute of Pathology, University of Bern, Switzerland) for providing CD1d/PBS-57-Tetramer. This study received financial support from the Swiss National Science Foundation (grants 182583 and 159188 to B.L., and 151370 to E.S.), the Deutsche Forschungsgemeinschaft (HE3116/9-1 to T.H. and B.L.). The funders had no role in the study design, data collection, and analysis, decision to publish, or preparation of the manuscript.

## Author contributions

B.L. designed the study, discussed data, and wrote the paper. H.-W.C. and U.M. performed experiments, analyzed data, and wrote the paper. L.O., M.N., C.E., C.G.-C., C.P.-S., M.L., and E.S. performed experiments, analyzed and discussed data. T.H. provided produced transgenic mice and discussed data.

## Competing interests

B.L., L.O., H.-W.C., C.G.-C., and C.P.-S. are cofounders and shareholders of Stromal Therapeutics. H.-W.C. is a part-time employee of Stromal Therapeutics. The remaining authors declare no competing interests.
