## [Peer Review File · Nature Communications]

Intestinal fibroblastic reticular cell niches control innate lymphoid cell homeostasis and functionThe previous round of reviews was completed at another journal

We would like to thank the Reviewers for the insightful comments. We have productively addressed the concerns of the reviewers, provide novel data and have amended our manuscript accordingly. Our responses to the reviewers' comments are listed in the following point-by-point reply.

Reviewer 1:

1. *The significance of the manuscript would be strengthened by studying the origin of SILT FRCs and the mechanisms involved in their differentiation pre-LTBR engagement. Differentiation trajectory analysis of scRNAseq data should provide insight into the origin of SILT FRCs and the mechanisms leading to their differentiation. I appreciate that the number of FRCs isolated is very small. However, are there any indications that FRCs from CP, imILF and mILFs differ?*

Based on our scRNA-seq data from adult intestinal fibroblasts, we have performed a differentiation trajectory analysis using the Slingshot algorithm. A distinct trajectory could be identified starting from CD81⁺ trophocytes to PDGFR α ^{lo} fibroblasts and subsequently bifurcating into two differentiation populations, i.e. SILT FRC and muscularis mucosae myocytes (Fig. 1R for the attention of the reviewer). In contrast, the trajectory analysis indicates that three separate cell clusters (PDGFR α ^{hi} telocytes, Thy1⁺ fibroblasts and mural cells) are not connected to the trophocyte-to-SILT FRC trajectory (Fig. R1). This result suggests that SILT FRCs may originate from CD81⁺ trophocytes in the adult small intestine. However, due to a lack of a suitable fate-mapping model, this pathway cannot be further validated.

Relation between FRCs in CPs, imILFs and mILFs: Indeed, cell numbers are too low to elaborate potential differences. The Ccl19-EYFP model would be suitable to enrich for SILT FRCs. We are planning to investigate in future studies the activation and differentiation of SILT FRCs under conditions of acute and chronic inflammation.

Figure R1. Differentiation trajectory analysis of adult intestinal fibroblasts. Slingshot pseudotime analysis of scRNA-seq data from adult intestinal fibroblasts. Left UMAP indicates different cell clusters. Right UMAP indicates predicted lineages using pseudotime inference of adult intestinal fibroblast differentiation.

2. *The data do not allow to conclude on the bi-directionality of the signals between SILT FRCs and ILCs, contrary to what is stated in the first paragraph of the discussion. How important is the provision of*

Lta by ILCs compared to B cells for example? *LTa* from B cells have been shown to be sufficient to drive ILF maturation.

While *IL-7* is very important for recruitment of ILCs, loss of ILCs in SILTs does not prevent the maturation of CP and imILF into mILFs when *Il7* is lost in *Ccl19+* FRC (Fig S7). Given the scRNAseq data generated in this study, could the authors evaluate the importance of some of the other interactions which may sustain FRC differentiation and ILC recruitment prior to *LTBR* engagement? *KIT-KITL* interaction seems like a good candidate and has been shown to be involved in Peyer's patches' formation (Chappaz, *Jl* 2010).

It has been shown previously that SILT development depends on the presence of *Rorc*-expressing lymphoid tissue inducer (LTi) cells and that the interaction with stromal cells depends on *LTβR* signalling (Tsuji et al., 2008, *Immunity*, PMID 18656387). Hence, LTi cells (= embryonic ILC3) initiate SILT development through the provision of *LTβR* ligands. In the adult, *LTβR* ligands are provided mainly by SILT B cells, which are important for ILF maturation (Hamada et al., 2002, *J Immunol*, PMID 11751946). We found that ILF maturation was not affected in SILTs of *Ccl19-Cre Il7^{fl/fl}* mice despite the reduction in ILC3 numbers. It is therefore conceivable - and in line with previous data - that both ILC3 and B cells provide *LTβR* ligands to promote ILF maturation.

We thank the reviewer for the suggestion to assess whether the *KIT-KITL* signaling affects ILC-FRC interaction. We found that ablation of *Kitl* in *Ccl19-Cre⁺* SILT FRCs did not alter ILC or lymphocyte numbers in the small intestine (Fig. R2a). Likewise, the development of Peyer's patches (Fig. R2b) or SILT structures (Fig. R2c) remained unaffected. This data suggest that *KITL* signaling in SILT FRCs is not critical for the process of SILT formation and maturation.

Figure R2. Genetic ablation of *Kitl* in SILT FRCs. **a**, Cell numbers of ILC subsets and *CD45⁺* cells, B cells and T cell subsets from adult *Ccl19-Cre Kitl^{fl/fl}* mice and co-housed littermate controls. **b**, PP number from adult *Ccl19-Cre Kitl^{fl/fl}* mice and co-housed littermate controls. **c**, Representative confocal microscopy images of SILTs from adult *Ccl19-EYFP Kitl^{fl/fl}* mice stained with the indicated antibodies. Scale bar, 20 μm, 20 μm and 50 μm respectively. (**a**) *n*=6 and 5 from 2 independent experiments, mean ± SEM (left panel) or geometric mean ± SD (right panel). (**b**) *n*=6 and 5 from 2 independent experiments, mean ± SEM. Statistical analyses were performed using non-parametric Mann-Whitney test (**a - b**)

3. The findings relative to *Citrobacter* infection have to be analysed and interpreted in the context of the loss of *Ltbr* in all FRCs not just SILT FRCs. In particular, the formation and function of Peyer's patches and mLNs will be affected and will probably impact the immune response to *Citrobacter rodentis*.

We appreciate the comment on the role of Peyer's patches and mesenteric lymph nodes during *Citrobacter* infection. Indeed, genetic *Ltbr* ablation in *Ccl19-Cre⁺* cells mice leads to impaired Peyer's patch development (Prados et al., 2020, Nat Immunol., PMID 33707780). However, as shown in Extended Data Figure 6, Peyer's patch development was not affected when *Ltbr* expression in adult *Ccl19-iEYFP Ltbr^{fl/fl}* mice was prevented through Doxycycline withdrawal. Likewise, development of mesenteric lymph nodes was not affected (Fig. R3a). Hence, adult *Ccl19-iEYFP Ltbr^{fl/fl}* mice (Dox 8 wk off) still possess Peyer's patches and mesenteric lymph nodes, but exhibit a loss of mILFs (Figure 4 of the manuscript). Moreover, B and T cell zone segregation remained unaltered (Fig. R3a) and we did not detect significant differences in the hematopoietic cell composition and ILC numbers from Peyer's patches or mesenteric lymph nodes in adult *Ccl19-iEYFP Ltbr^{fl/fl}* mice after withdrawal of the Doxycycline treatment for 8 weeks (Fig. R3b). Taken together, these results provide further evidence that it is the $LT\beta R$ signaling in SILT FRCs of adult mice that affects the local immune response against *Citrobacter rodentium*.

Figure R3. Transgene activity and hematopoietic cell numbers in *Ccl19-iEYFP Ltbr^{fl/fl}* mice after withdrawal the doxycycline treatment for 8 weeks. **a, Representative confocal microscopy images of mLN and PP structures stained with indicated markers. Scale bar, 300 μ m and 50 μ m (mLN), 150 μ m and 50 μ m (PP). **b**, Cell numbers of CD45⁺ cells, B cells and T cell subsets (left panels) and ILC subsets (right panels) from adult *Ccl19-iEYFP Ltbr^{fl/fl}* mice and co-housed littermate controls after withdrawal of the doxycycline treatment for 8 weeks. (**b**) $n=9$ and 12 from 3 independent experiments, geometric mean \pm SD. Statistical analyses were performed using and non-parametric Mann-Whitney test.**

Minor comments:

- Could you clarify how cells were sorted for the scRNAseq analysis?

The sorting markers have been added in the legend of Figure 1a.

- It would be informative to have a full list of DEGs for SILT FRCs.

A full list of differentially expressed marker gene is now provided in the Extended Data Table 1 and mentioned in the revised manuscript.

- What is the level of expression of *Ltbr* across SI fibroblast populations?

The *Ltbr* gene is expressed by all SI fibroblasts but slightly enriched in SILT FRC (Fig. R4).

Figure R4. *Ltbr* expression in adult intestinal fibroblasts. Violin plots of *Ltbr* expression based on the scRNA-seq analysis of adult intestinal fibroblasts.

- Figure 5d: is the legend right regarding the opacity of the circles?

Yes, the opacity indicates the p values derived from the Cellphone analysis.

Reviewer 2:

Overall, this is a technically well-performed study, provides convincing results and a clear message. However, my excitement is limited because this study doesn't provide novel insights but instead redemonstrates well established findings in a technically refined manner. This refers to the role of LTb with regards to FRC functionality in general and specifically for CP and ILFs (Taylor et al. 2004, PMID: 15585839). This paper needs to be cited and discussed. In 2007 paper, Taylor et al. further characterized the stromal populations that receive LTb signals and established that LTb is continuously required for ILF maintenance (PMID: 17442949). Also, it is well established that FRC provide a critical source for IL7 (PMID: 17893676) and that in turn IL7 together with IL15 are critical for ILC maintenance and survival in the intestine (PMID: 28361874). The work by Robinette et al. as well as other studies further established the link between ILC abundance in the intestine and resistance towards C. rodentium infections. Therefore, I don't see how the work by Cheng et al provide new insights or a new concept that would merit publication in Nature Immunology.

We have taken into consideration the comments of Reviewer 2 concerning previously published studies that have used globally gene-deficient animals for the characterization of FRC-ILC interaction. As highlighted in the discussion, cell type-specific and timely regulated ablation of specific genes does not merely “refine” previously published work, but pinpoints specific cellular and molecular interactions that are of biological importance.

Reviewer 3:

1. Although *CCL19*+ FRCs underly gut lymphoid structures, ILCs are distributed throughout the gut tissue. Even ILC3s themselves have subsets that are associated with these lymphoid structures (*CCR6*+ enriched) but also ILC3s present more diffusely through the lamina propria (*NCR* enriched). As such, the fundamental question of how localized lymphoid tissue structures impact distributed ILCs remains unclear. Do the FRCs act via ILC precursor subsets? Or are these *CCL19*+ FRCs actually more broadly distributed across the SI? Or are ILC3s really preferentially impacted, particularly after inflammation?

We thank the Reviewer for the productive comments. We agree that SILT FRCs may indeed act on ILC precursor cells (ILCp). However, the current animal models or markers, such as *Id2* or *Zbtb16* are limited in their ability to label ILC progenitors in the bone marrow. In addition, these reporter genes broadly target ILCs and even different lymphocyte subsets in other organs (Xu et al., 2019, Immunity, PMID 30926235). Therefore, we feel that it is currently not feasible to properly identify and locate ILC progenitors in the adult small intestine.

The *Ccl19*-EYFP model highlights FRCs in SILTs as shown in Figure 1b-f. Since more than 90% of small intestinal ILC3 reside in SILTs (Pearson et al., 2016, eLife., PMID 26780670) and the ILC1 population is to some extent derived from *Rorc*+ ILC3 (ILC1/exILC3) (Vonarbourg et al., 2010, Immunity., PMID 21093318; Fiancette et al. 2021, Nat Immunol., PMID 34556884) it is conceivable that SILT FRCs interact preferentially with ILC3.

To address the question whether ILC3 are preferentially affected during inflammation, we infected *Ccl19*-Cre *Ltbr*^{fl/fl} mice and littermate controls with *Citrobacter rodentium* and evaluated lymphocyte and ILC changes on day 6 after infection. This analysis did not reveal significant differences between *Ccl19*-Cre *Ltbr*^{fl/fl} mice and cohoused littermate controls after infection (Fig. R5a). However, total CD127⁺ ILC and ILC3 numbers were significantly reduced in *Ccl19*-Cre *Ltbr*^{fl/fl} mice after bacterial infection, whereas ILC1 and ILC2 numbers showed only a slight reduction under conditions of FRC-specific *Ltbr* deficiency (Fig. R5b). Taken together, these results indicate that the cellular niche formed by SILT FRCs is particularly important for ILC3.

Ccl19-Cre⁺ FRCs in the SI lamina propria are located in SILTs, as highlighted in the manuscript and shown here again for the attention of the Reviewer in Figure R6.

Figure R5. Lymphocyte and ILC numbers from the *Ccl19*-Cre *Ltbr*^{fl/fl} mice and cohoused littermate controls after *Citrobacter rodentium* infection. Cell numbers of T cell, T cell subsets and B cells (a) or ILC subsets (b) from the small intestinal lamina propria of *Ccl19*-Cre *Ltbr*^{fl/fl} mice and co-housed littermate controls on day 6 after *Citrobacter* infection. (a) $n=8$ and 8 from 3 independent experiments, geometric mean \pm SD. (b) $n = 8$ and 8 from 3 independent experiments, mean \pm SEM. Statistical analyses were performed using non-parametric Mann-Whitney test (a) or unpaired student *t* test (b).

2. *The impact of CCL19+ FRCs in lymph nodes and Peyer's patches could be significant and is not discussed or tested here. As such, the specificity of the approach for intestinal FRCs (and intestinal ILCs) is not demonstrated. In a related point, the actual contribution of intestinal ILCs (and not other intestinal lymphoid subsets) to the colitis phenotype is not demonstrated and would strengthen the author's most important contention, that CCL19+ FRCs specifically support ILCs.*

To address this question, we have evaluated the structural integrity and hematopoietic cell composition in the PPs and mLN from *Ccl19-iEYFP Ltbr^{fl/fl}* mice (Dox 8 wk off). Please see our response to Reviewer 1 point 3.

3. *The authors find that inducible loss of IL-7 in CCL19+ cells lead to fewer of all intestinal ILCs, including fewer Rorgt+ cells in ILFs (but not CPs). However, these mice are not well explored. What is phenotype after colitis? Does this impact intestinal (comparing to SLO) T-cell, innate-like T-cell, and ILC composition and function at rest or after inflammation?*

To address this point, we have further characterized T cell subset composition, including conventional CD4 and CD8 T cells, Th17 cells and Treg cells, as well as innate T cell subsets such as $\gamma\delta$ T cells and CD1d-tetramer⁺ iNKT cells. Conditional ablation of *Il7* gene in SILT FRCs did not affect T cell subset composition in the lamina propria of *Ccl19-Cre Il7^{fl/fl}* mice compared to co-housed littermate controls (new Extended Data Fig. 7e-g). The infection with *Citrobacter rodentium* in these mice revealed shortened colon length and increased bacterial concentration in the faeces and colon (new Extended Data Fig. 7h-i). These data further underpin the conclusion that SILT FRCs provide critical niche factors that support ILC sustenance and function.

Minor comments:

-Line 54-55 states "parabiosis and cell fate-mapping experiments in mice have confirmed replenishment of ILCs from circulating ILCp and revealed ILCp recruitment during inflammation". I believe these studies more accurately show that tissue ILCs can be replenished by ILCp and that blood ILCp OR mature ILCs contribute to tissue pools, particularly during inflammation. For example, Ron Germain's initial work in Science (authors reference 14) shows that intestinal 'mature' ILC2s can migrate to lung and contribute to anti-helminth response. Suggest rewording of this section, particularly since the author's do not directly test ILCp versus mature ILCs in their studies.

-Line 56 – see above, tissue ILCs can come from tissue resident mature ILC proliferation, ILCp maturation (in tissue or from circulation), or trafficked mature ILCs from circulation. The relative contribution of each of these pathways is not definitively established, although proliferation of tissue-resident mature ILC pools is certainly a major one...

-I do not believe the authors ever define for the reader the definitions and/or differences between SILTs, CPs, immature ILFs, and mature ILFs. Brief description in intro is warranted.

In Figure 1 and supplemental, the authors describe a cluster as vascular smooth muscle cells. However, these cells seem to be more consistent with 'pericytes', often defined by the exact markers shown by the authors...perhaps a more inclusive term would be 'mural' cells to avoid the nebulous distinction between pericytes and vascular smooth muscle cells using scRNAseq.

We have amended the text and the figures according to the reviewer's comments in the revised version.

Although it is clear that Ccl19+ FRCs are enriched in SILT structures, more definitive imaging using the lineage tracker and CLU staining would be required to show they are NOT present outside of these structures. This seems critical, as the ultimate phenotypes ascribed to these CCL19+ intestinal FRCs include impacts on ILC1s, ILC2s, and ILC3s, but only ILC3 (subsets) are known to be enriched around SILTs. Authors could consider analysis of larger image volumes, for example.

We have thoroughly analyzed the transgene activity in the small intestine. The Ccl19-EYFP transgene almost exclusively detectable in FRC of Peyer's patches (Prados et al., 2021, Nat Immunol., PMID 33707780) and SILTs (Fig. R6 and Fig. 1b-c). Likewise, CLU expression is highly expressed FRCs of Peyer's patches (Prados et al., 2021, Nat Immunol., PMID 33707780) and SILTs (Fig. R6 and Extended Fig. 1e).

Figure R6. *Ccl19-EYFP transgene labels SILT FRC in the small intestine. Representative confocal microscopy images of Ccl19-EYFP transgene activity in small intestine stained with indicated markers. Boxed areas indicate the magnified regions shown in right panels. Scale bar, 200 μm, 50 μm and 50 μm.*

Fig 2D – scRNAseq of control and Ltbr floxed intestinal FRCs seem to show shifts in potential FRC subclusters (but not all subclusters), further comments or analysis is warranted.

Indeed, the shift of cell clusters in presented UMAPs represents differential gene expression between subsets/subclusters. *Ltbr*-deficient FRCs underpinning CP and immature ILF exhibit the expression of conventional SILT FRC markers such as *Cxcl16* and *Kitl*, but show lower expression of some chemokine and cytokine genes (Fig. 2e). The *Ltbr*-dependent shift in gene expression/phenotype founds the basis for the impediment of ILF maturation.

-Given the role of Th17s and Tregs in gut homeostasis and response to pathogens, a specific analysis of the impacts on these two subsets is warranted in the Ltbr floxed mice (not just bulk CD4 T cells). This would also help to interpret the Citrobacter infection data, where Th17s and ILC3s can cooperate to limit restrict infection. For example, depleting T-cells or blocking T-cell trafficking during infection may help clarify that ILCs, and not other lymphocytes, are driving this phenotype. In a broader sense, other intestinal innate-like T-cells (gamma deltas, MAITS, etc) could also be impacted in this experimental setup and are not directly examined.

To address this question, we have analyzed the composition of innate T cells including $\gamma\delta$ T cells and iNKT cells in Ccl19-Cre *Ltbr*^{fl/fl} mice and co-housed littermate controls. In sum, we

conditional ablation of the *Ltbr* gene in SILT FRCs did not affect T cell subset composition in the lamina propria (Fig. R7). Since only few T cells can be detected in SILTs, it is conceivable that an altered SILT maturation has no or only very limited effect on the immune functions T cell subsets in the small intestine.

Fig. R7. T cell subset composition in the small intestine of *Ccl19-Cre Ltbr^{fl/fl}* mice and cohoused littermate controls. *a.* Representative gating strategy of T cell subsets isolated from the small intestine of *Ccl19-Cre Ltbr^{fl/fl}* mice and cohoused littermate controls. *b.* Cell number of $\gamma\delta$ T cells, iNKT cells and Treg from naïve *Ccl19-Cre Ltbr^{fl/fl}* mice and cohoused littermate controls. *c.* Cell number of IL-17⁺ T cells isolated from *Ccl19-Cre Ltbr^{fl/fl}* mice and cohoused littermate controls under naïve condition. (*b-c*) $n=5$ and 4 from 2 independent experiments, geometric mean \pm SD. Statistical analyses were performed using non-parametric Mann-Whitney test (*b - c*)

-With inducible loss of FRC-*Ltbr*, was there impacts on relevant T-cell subsets?

We have evaluated the B and T cell number from *Ccl19-iEYFP Ltbr^{fl/fl}* mice and co-housed littermate controls after withdrawal of Doxycycline for 8 weeks (Dox off 8 wk). There could not detect no significant differences in either B or T cell numbers (Fig. R8).

Fig. R8. B and T cell number in the small intestine of *Ccl19-iEYFP Ltbr^{fl/fl}* mice and cohoused littermate controls after the withdrawal of doxycycline for 8 weeks. Cell numbers of CD45⁺ cells, CD19⁺ B cells and T cell subsets from intestinal lamina propria of *Ccl19-iEYFP Ltbr^{fl/fl}* mice and co-housed littermate controls after the removal of dox for 8 weeks. $n=4$ and 8 from 2 independent experiments, geometric mean \pm SD. Statistical analyses were performed using non-parametric Mann-Whitney test.

The authors emphasize the FRC-impacts on ILC3s, but always see impacts on ILC1s and variably reach significance for decreased ILC2s. As such, their conclusions would be strengthened with comparisons to other intestinal inflammatory models (helminth infection, intracellular bacteria/virus) that could further test the relevance of impacts on ILC1s and ILC2s. Any results would be meaningful here and might suggest that the FRC-mediated decline in ILC3s is most meaningful, which would fit with the author's imaging data. This should at least be discussed in the work.

To address this question, we have infected Ccl19-Cre *Ltbr*^{fl/fl} mice and cohoused littermate controls with L3 larvae of *Heligmosomoides polygyrus* helminths. In line with the fact that neither ILC2 nor eosinophils were altered in the deficiency of LTβR signalling in SILT FRC (Fig. 3c and Extended Fig. 4d), there was no significant difference in the granuloma formation or worm burden in the small intestine on day 8 after helminth infection (Fig. R9a-b). In contrast, conditional ablation of *Ltbr* gene in SILT FRCs resulted in the reduction of ILC3 cell numbers in naive mice (Fig. 3c and Fig. 4d) and after *Citrobacter rodentium* infection (Fig. R5b). Taken together, these results support the notion that the interaction between SILT FRCs and ILC3 mainly promotes intestinal immune responses against bacterial infection. Certainly, further studies are required to fully elaborate the extent of SILT FRC-ILC2 interaction in type 2 immunity against parasite infection.

Figure R9. Granuloma number and worm burden in the small intestine of Ccl19-Cre *Ltbr*^{fl/fl} mice and cohoused littermate controls on day 8 after *Heligmosomoides polygyrus* infection. a-b, Granuloma number and worm burden from the small intestine of Ccl19-iEYFP *Ltbr*^{fl/fl} mice and co-housed littermate controls on day 8 after helminth infection. n=4 and 3, mean ± SEM. Statistical analyses were performed using non-parametric Mann-Whitney test.

REVIEWERS' COMMENTS

Reviewer #1 (Remarks to the Author):

I thank the reviewers for their responses to my concerns. I feel the current work is strengthened, accurate, and impactful and do not have further comments.

Reviewer #2 (Remarks to the Author):

I would like to thank the authors for carefully answering my comments. This manuscript provides an elegant characterisation of intestinal FRCs and their importance in the control of ILCs which will be of interest to the readers of Nature Communications. I have no further comments.